# Direct generation of protein conformational ensembles via machine learning

Giacomo Janson[1], Gilberto Valdes-Garcia [1], Lim Heo [1] & Michael Feig [1] ✉

Dynamics and conformational sampling are essential for linking protein structure to biological function. While challenging to probe experimentally, computer simulations are widely used to describe protein dynamics, but at significant computational costs that continue to limit the systems that can be studied. Here, we demonstrate that machine learning can be trained with simulation data to directly generate physically realistic conformational ensembles of proteins without the need for any sampling and at negligible computational cost. As a proof-of-principle we train a generative adversarial network based on a transformer architecture with self-attention on coarse-grained simulations of intrinsically disordered peptides. The resulting model, idpGAN, can predict sequence-dependent coarse-grained ensembles for sequences that are not present in the training set demonstrating that transferability can be achieved beyond the limited training data. We also retrain idpGAN on atomistic simulation data to show that the approach can be extended in principle to higher-resolution conformational ensemble generation.

The biological function of a protein is determined not just by a single three-dimensional (3D) structure but by its dynamical properties that give rise to conformational ensembles[1]. Characterizing conformational ensembles is crucial to mechanistically understand the activity of proteins and their regulation, and impacts biomedical sciences, biotechnology and drug design[2–4].

Experimental techniques for probing the dynamics of biomolecules suffer from low spatial or temporal resolution[5]. Alternatively, computational methods are employed to investigate protein dynamics and generate structural ensembles. A powerful strategy is the use of physics-based molecular dynamics (MD) simulations[6]. MD samples from possible configurations of a molecular system to identify the energetically most favorable regions in conformational space. However, high dimensionality and significant kinetic barriers result in formidable computational challenges for all but the very simplest protein systems, even with specialized computer hardware[7] or enhanced sampling methods[8]. New strategies for accelerating the generation of biologically relevant dynamic ensembles are thus needed.

In recent years, data-driven machine learning techniques have successfully tackled the challenge of predicting the 3D conformation of a protein given its amino acid sequence[9]. Machine learning methods, such as AlphaFold 2 (AF2)[10,11], reach remarkable accuracy compared to experimentally-determined structures[12]. However, these methods do not reflect that proteins are dynamic entities with multiple conformational states. This is especially true for intrinsically disordered proteins (IDPs)[13], which lack stable structures and exhibit high conformational variability[14,15].

Machine learning methods are also a promising strategy for accelerating the generation of protein dynamics and conformational ensembles. Multiple strategies have been explored to facilitate the analysis of complex molecular simulations[16], to guide MD via enhanced sampling techniques[17,18], or to provide optimized energy functions[19]. Another strategy, followed in this work, is to directly model conformational ensembles through generative models[20]. Generative models are based on neural networks and have been effective in several artificial intelligence tasks[21,22]. For generating conformational ensembles, such models may be trained on datasets of molecular conformations obtained by "classical" computational methods, such as MD. The idea is essentially to learn the probability distribution of the conformations from a training set. Once trained, such models can be

---

[1]Department of Biochemistry and Molecular Biology, Michigan State University, East Lansing, MI 48824, USA. ✉e-mail: mfeiglab@gmail.com

used to quickly draw statistically-independent samples from these complex, highly-dimensional distributions[23,24]. Because generative models are not subject to kinetic barriers, they circumvent the main bottleneck of MD sampling.

For generative models to have real utility, it is essential that previously unseen molecular conformations can be generated for a given system, and that conformations can be generated for new systems with different chemical compositions from systems in the training set. This can be achieved with conditional generative models that are trained with data from multiple molecules by using their composition as conditional information. If the training data is large enough, such models are expected to learn not just the probabilities of training conformations, but to learn transferable features of how favorable conformations are constructed. Conditional generative models have been applied so far only on small molecules[25–27]. While there have been advances in the unconditional modeling for proteins[28], conditional modeling for these complex molecules has not been explored yet. Developing an accurate conditional model for proteins would allow direct and computationally efficient generation of conformational ensembles for any protein sequence.

In this study, we present a conditional generative model for the generation of protein conformations trained on molecular mechanics simulation data. We apply it here to model the ensembles of IDPs. We chose to work with IDPs because of their conformational variability. Given the complexity of the problem, we apply a simplified description based on a Cα representation. Training data consists of MD simulations of IDPs obtained using a residue-level CG force field (FF), all-atom implicit solvent simulations with the ABSINTH potential[29], and the conformational ensemble of α-synuclein from all-atom explicit solvent simulations[30]. We use a Generative Adversarial Network (GAN)[31] to learn the distribution of 3D conformations in the reference datasets. GANs are generative models that have previously been employed in molecular modeling tasks[32,33]. Our model, called idpGAN, has a network architecture that incorporates ideas from machine learning models used in protein structure prediction[10]. It directly outputs 3D Cartesian coordinates and can model previously unseen

conformations of variable sequences and lengths. Since GANs have fast sampling capabilities, our model can generate thousands of independent conformations in fractions of a second, providing a computationally efficient way to reproduce conformational ensembles.

## Results

### IdpGAN network architecture and training

IdpGAN is a generative model trained on MD data to directly output 3D molecular conformations at a Cα coarse-grained level. From different types of generative models, we chose here GANs[31] because of their ability to generate high-quality samples and their fast sampling capabilities[20]. The learning process of a GAN involves an adversarial game between two neural networks, a generator (G) and a discriminator (D) (Fig. 1). The G network of idpGAN is based on a transformer architecture[34] (Supplementary Fig. 1). When generating a conformation for a protein with $L$ residues, a latent sequence $\mathbf{z} \in \mathbb{R}^{L \times n_z}$ is sampled (its values are randomly extracted from a normal prior). The G network takes as input $\mathbf{z}$, progressively updates it through a series of transformer blocks and outputs a sequence $\mathbf{r} \in \mathbb{R}^{L \times 3}$ corresponding to the 3D coordinates of the Cα atoms of the protein. In addition to $\mathbf{z}$, the network also takes as input the amino acid sequence $\mathbf{a} \in \mathbb{R}^{L \times 20}$ with one-hot encodings. The sequence provides the conditional information for modeling proteins with different amino acids. Transformer-like architectures are the cornerstone of AF2[10], and they are well-suited for protein conformation generation. They allow variable-size outputs so that proteins of different lengths can be modeled. Additionally, a self-attention mechanism ensures that each of the $L$ tokens in an embedding sequence (corresponding to residue representations) are updated using information from the rest of the sequence, thus helping to form consistent 3D structures[9].

In GAN training, the role of the D network is to drive G to generate data distributed like in the training set. In idpGAN, D receives as input a protein conformation $\mathbf{x}$ and an amino acid sequence $\mathbf{a}$. It returns a scalar value corresponding to the probability of the combination being real (according to the training set). The input $\mathbf{x}$ contains the values of

**Train D on generated data: classify data as fake. Train G: make D classify generated data as real.**

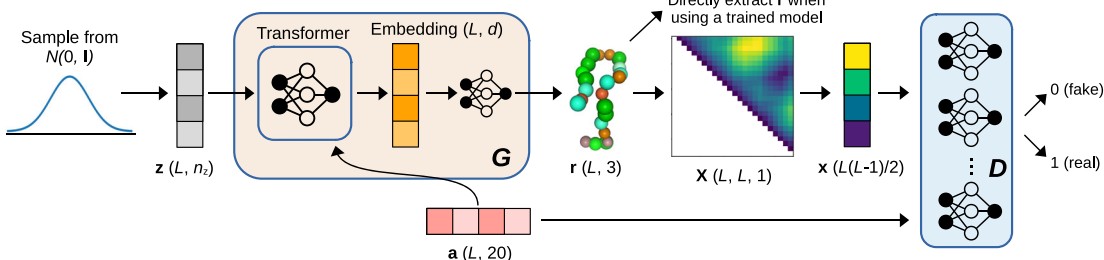

**Train D on real MD data: classify data as real.**

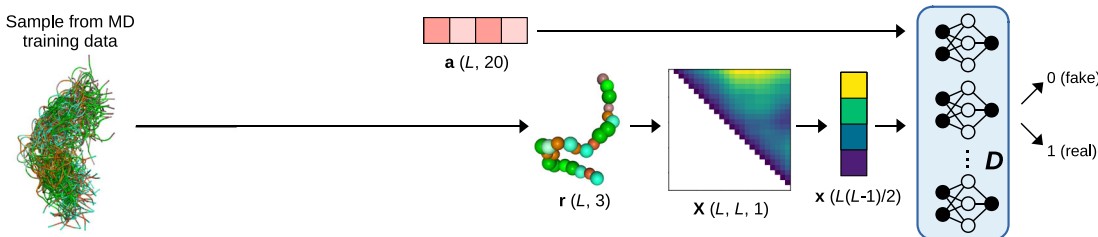

**Fig. 1 | Overview of the idpGAN network architecture.** A $\mathbf{z}$ latent sequence is used as input to the G network. Amino acid information $\mathbf{a}$ is also provided as input to G. The output of the last transformer block of G is mapped to 3D Cartesian coordinates $\mathbf{r}$ through a position-wise fully-connected network. Conformations $\mathbf{r}$ from G and the training set are converted in distance matrices and their upper triangles are used as input to a set of D networks, which also receive as input $\mathbf{a}$. The objective of the D networks is to correctly classify real (MD) and fake (generated) samples. The objective of G is to generate increasingly realistic samples to decrease the performance of D. Once idpGAN is trained, the generated coordinates $\mathbf{r}$ from G can be directly used without converting them to distance matrices. Molecular structures were rendered with NGLview[68].

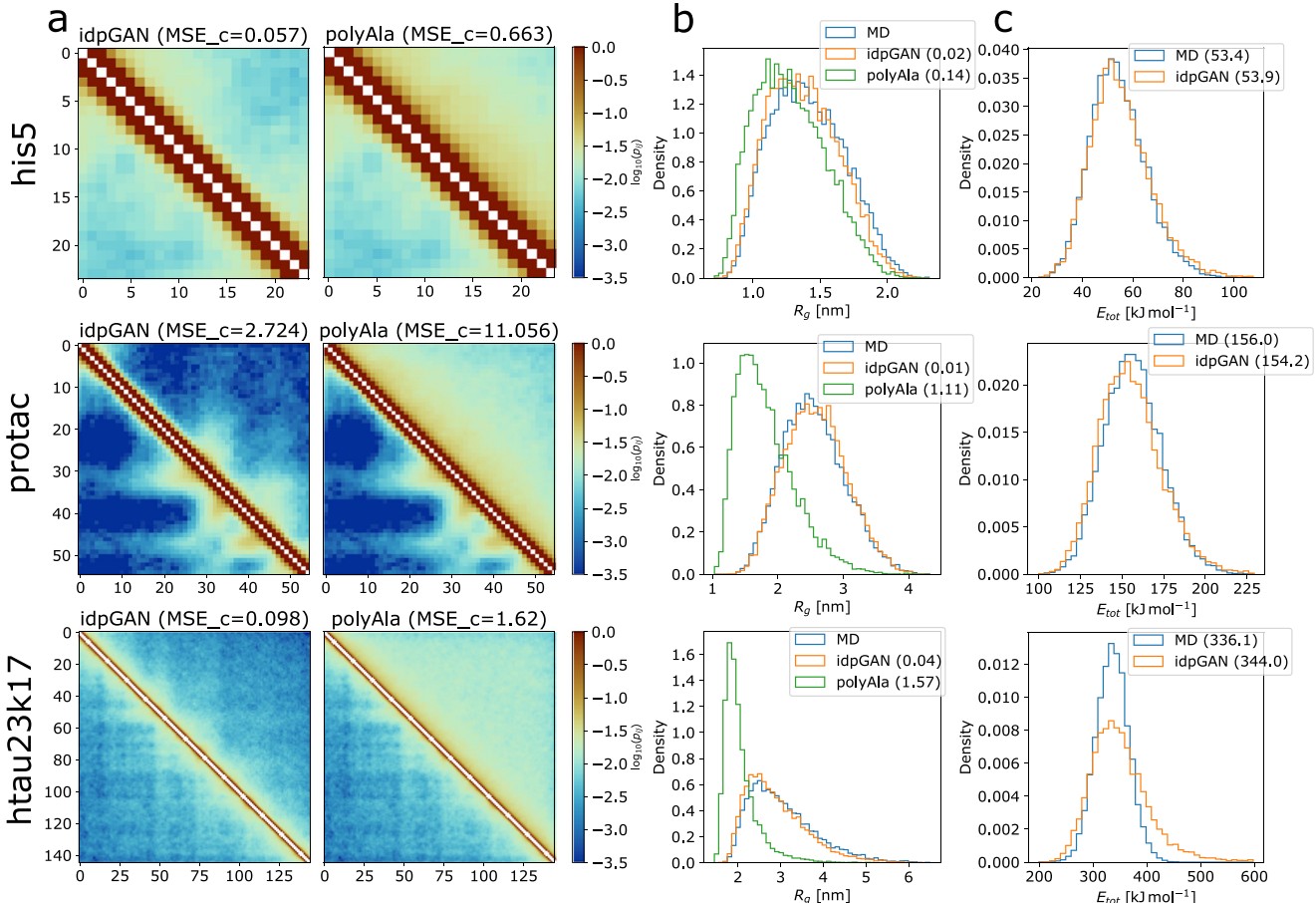

**Fig. 2 | Examples from idpGAN for three IDP_test proteins.** Each row of the image shows data for his5 ($L = 24$), protac ($L = 55$), and htau23k17 ($L = 145$), respectively. For each protein, the following information is reported: **a** idpGAN (left) and polyAla (right) contact maps are shown in the upper triangles of the images. The MD maps are shown in the lower triangles. The values shown along the horizontal and vertical axes are residue indices. Each cell of a map represents a residue pair and is colored according to its $\log(p_{ij})$ value, where $p_{ij}$ is the probability of observing a contact between residues $i$ and $j$ in an ensemble (see "Methods" for the definition of a contact). The MSE_c scores with the MD maps are shown in brackets. **b** Radius-of-gyration distributions for the MD (blue), idpGAN (orange) and polyAla (green) ensembles. KLD_r (cf. "Methods") values are shown in brackets. **c** Total potential energy distributions for the MD and idpGAN ensembles based on the CG energy function. The median values of the data are shown in brackets.

the interatomic distance matrix calculated from coordinates **r**. Since interatomic distances are E(3) invariant with respect to transformations of atomic coordinates, using **x** as input to D makes idpGAN training invariant to translations, rotations, and reflections of the input conformations, an important requirement in 3D molecular generative models[26]. When modeling CG MD data, we can allow reflection invariance in our model since the residue-based protein representation that we use is not chiral. When modeling all-atom datasets, we adopted different strategies to handle chirality (cf. "Methods"). To train idpGAN, the D network must process data from proteins with variable lengths. Although we experimented with different network architectures receiving inputs of variable sizes[35], we could not identify a solution resulting in stable GAN training. Instead, for the model trained on CG MD data, a simple multilayer perceptron (MLP) gave good results. Since MLPs take fixed-size input, we employed four MLPs discriminators (each accepting data from proteins with a certain length) along with a scheme for randomly cropping conformations (so that all training proteins could be accepted by one of the MLPs). The idea of using one G and multiple D networks in GANs follows previous works[36,37]. When training on all-atom ABSINTH data, we employed a single 2D convolutional network discriminator that uses zero-padding to account for different input sizes.

IdpGAN was trained on conformations from simulations of large sets of IDPs with lengths ranging from 20 to 200 residues.

Simulations were performed using either a residue-level (Cα-based) CG model or all-atom models. When building the training sets, our aim was to span a significant portion of the IDP sequence space, so that our models could learn general rules relating to sequences and conformational variability that can be transferred to new sequences.

## IdpGAN based on CG simulations

We initially trained idpGAN on MD data obtained with a residue-based protein model. Since idpGAN is a conditional generative model, it can generate conformations for proteins of arbitrary sequences. After training, we evaluated the model on CG MD data for a set of 31 selected IDPs, named IDP_test, that are not in the training set and have no similar sequences in it (cf. "Methods"). We also compared against data from CG MD simulations for poly-alanine (polyAla) chains with lengths from 20 to 200 as a random linear polymer without sequence-specific interactions to study whether idpGAN could provide better approximations.

Examples of generated ensembles for three selected IDP_test proteins (his5, protac and htau23k17) are shown in Fig. 2 (the remaining proteins are shown in Supplementary Figs. 2–5). Sample illustrations of generated 3D conformations along with their nearest neighbor in the MD data (Supplementary Fig. 6) demonstrate that the conformations appear qualitatively "realistic" and capture diverse

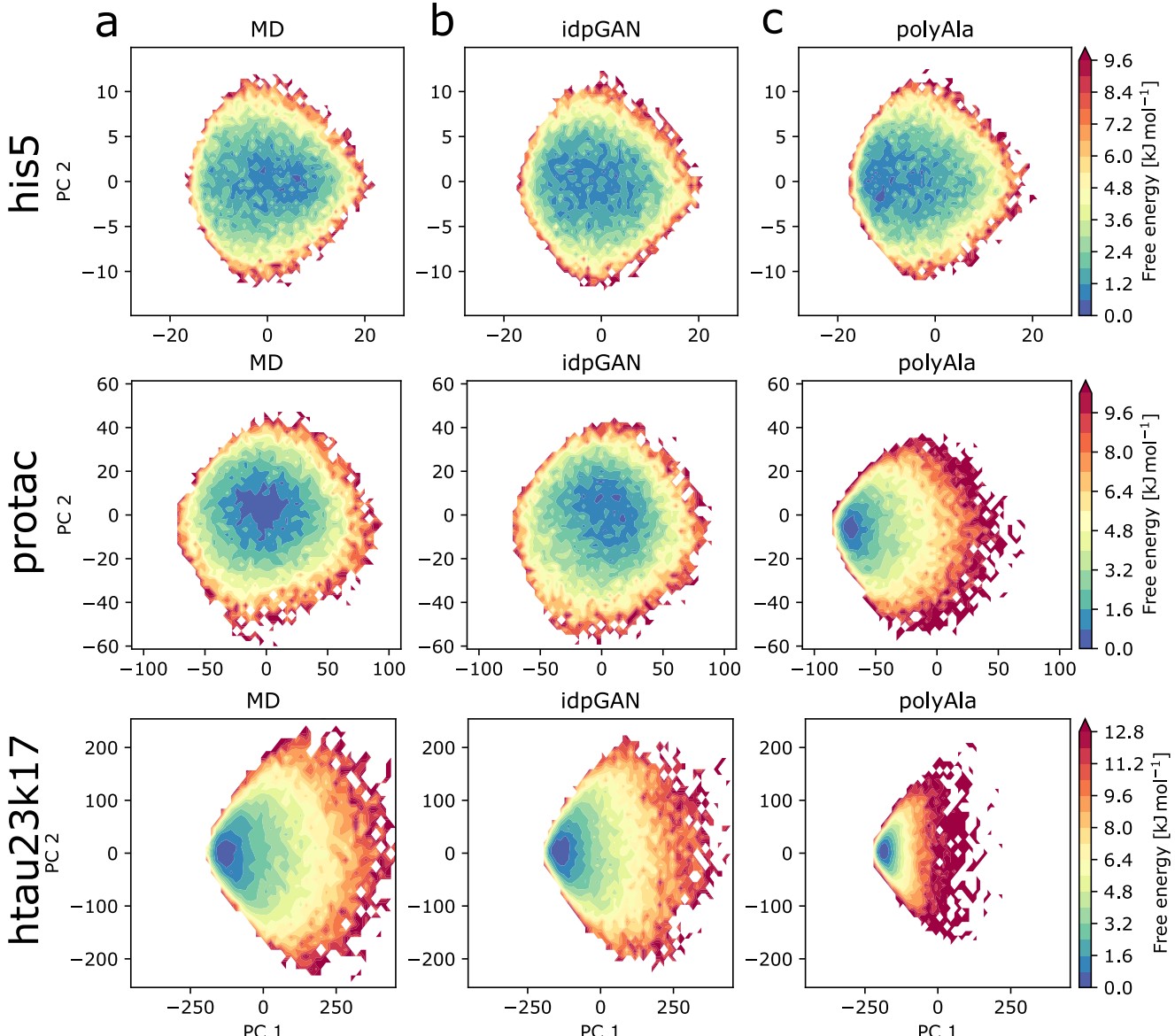

**Fig. 3 | Potential of mean force in PCA space.** The figure shows the potential of mean force profiles of MD (reference) (**a**), idpGAN (**b**) and polyAla (**c**) ensembles of the his5, protac, and htau23k17 proteins from the IDP_test set. For each protein, the conformations from the ensembles are projected along the first two components of a PCA model fitted on reference MD data.

types of structures found in the MD data of the IDP_test proteins. Ensemble properties such as residue contact maps are also matched closely in the examples. In the CG model that we study, specific amino acid sequences influence protein dynamics and give origin to contact maps with patches of relatively lower or higher contact probabilities. The goal of idpGAN was to capture these sequence-specific patterns. In the case of protac, there are low-probability regions in the contact map caused by stretches of negatively charged amino acids repelling each other (Supplementary Table 1). The map generated by idpGAN reproduces a very similar distinctive pattern, even though protac (or a similar IDP) was not present in the training set. This clearly illustrates that idpGAN learned transferable residue-specific interaction patterns from the training MD data. Finally, Fig. 2 also shows radius-of-gyration and energy distributions (based on the CG model energy function) from the idpGAN-generated models in good agreement with the MD-generated ensembles. This indicates that the models are of practical value in estimating radius-of-gyration distributions and that they are of high structural quality without clashes or other significant violations of stereochemical constraints.

Potentials of mean force based on principal component analysis (PCA) were analyzed to further characterize the ensembles generated by idpGAN as in related work[38]. We first performed PCA on the reference MD data using interatomic distances as input features. We then projected the conformations from idpGAN onto the PCA space defined by the MD data. For control, we also projected snapshots from the MD simulations of polyAla chains with the same number of amino acids. In Fig. 3, we plot the free energy landscapes along the first two principal components for the three selected IDP_test set proteins (other proteins in Supplementary Figs. 2–5). We find that idpGAN provides good approximations of the MD ensembles, whereas polyAla ensembles show clear differences, especially for protac and htau23k17.

Different metrics were evaluated across all IDP_test proteins to quantitatively evaluate idpGAN (Fig. 4 and Table 1). Values closer to zero reflect better approximations of the MD reference ensembles. The same metrics were also calculated for the corresponding polyAla ensembles to provide a random polymer baseline. Another baseline was calculated by drawing snapshots from an additional independent long MD run and comparing those to the MD snapshots from the same

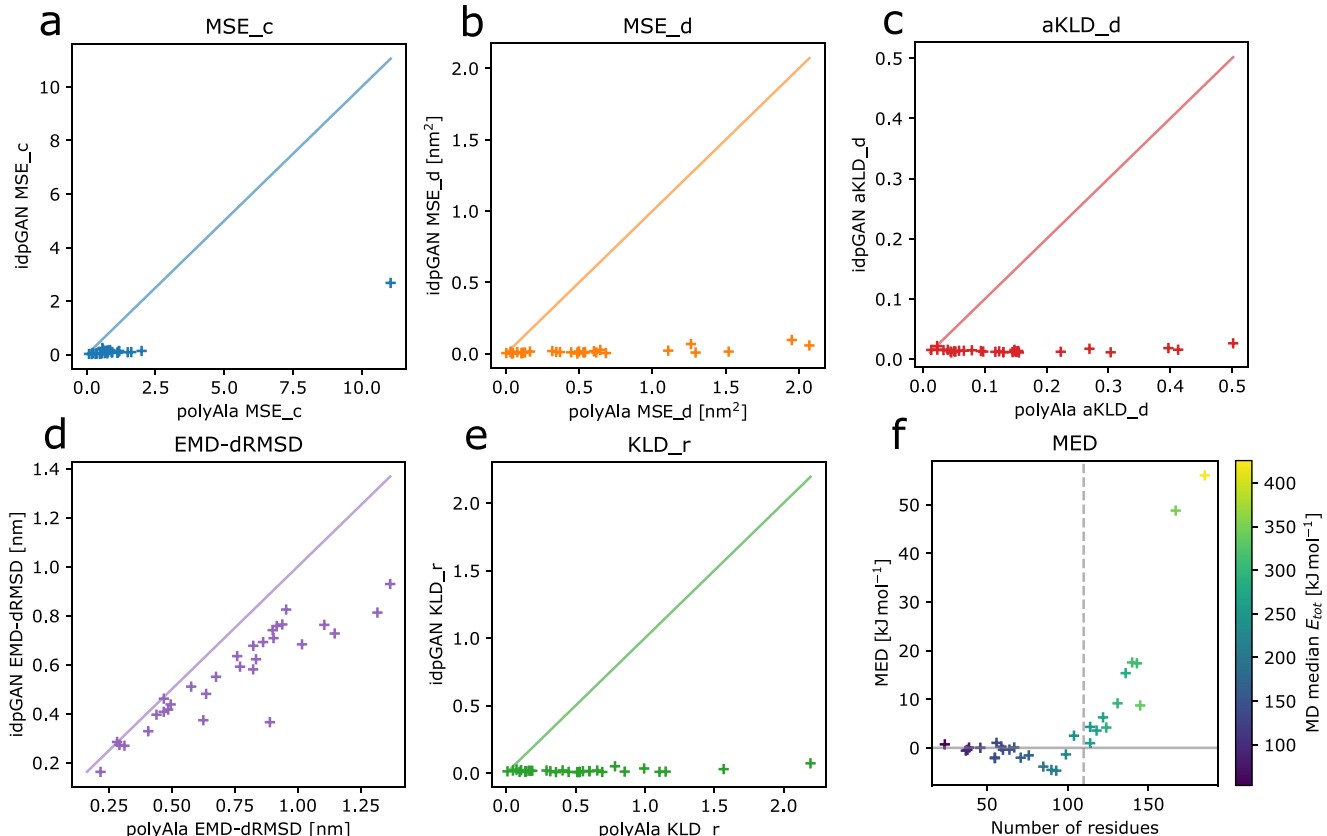

**Fig. 4 | Evaluation of idpGAN and polyAla ensembles for approximating MD data.** Results are reported for IDP_test set proteins ($n = 31$). **a**, **b**, **c**, **d**, and **e** show the values of MSE_c, MSE_d, aKLD_d, EMD-dRMSD and KLD_r, respectively, obtained by polyAla (x-axis) and idpGAN (y-axis) for all the proteins in the set. See "Methods" for a description of the metrics. Lower values indicate better performance in approximating MD ensembles. IdpGAN MED values as a function of protein length are shown in **f** with markers colored according to the median potential energy of proteins in the MD ensembles. The dashed vertical line represents the maximum crop length used in idpGAN training ($L = 110$). Source data are provided as a Source Data file.

### Table 1 | Evaluation of idpGAN for the IDP_test set

| Method | MSE_c | MSE_d [nm²] | aKLD_d | EMD-dRMSD [nm] | KLD_r | MED [kJ mol⁻¹] |
|---|---|---|---|---|---|---|
| polyAla | 1.10 ± 0.34 | 0.55 ± 0.10 | 0.14 ± 0.02 | 0.73 ± 0.05 | 0.53 ± 0.09 | - |
| idpGAN | 0.18 ± 0.08 | 0.02 ± 0.004 | 0.01 ± 0.001 | 0.56 ± 0.04 | 0.02 ± 0.002 | 5.58 ± 2.48 |
| null-MDᵃ | 0.09 ± 0.06 | 0.0003 ± 0.00004 | 0.01 ± 0.00005 | 0.54 ± 0.03 | 0.01 ± 0.0003 | −0.08 ± 0.10 |
| idpGAN no-stereoᵇ | 0.22 ± 0.09 | 0.01 ± 0.003 | 0.01 ± 0.001 | 0.55 ± 0.04 | 0.01 ± 0.001 | 67410 ± 34340 |

Average values are reported along with standard errors for all the proteins in the set ($n = 31$). Source data are provided as a Source Data file.
ᵃindependent MD run compared with the reference MD runs.
ᵇidpGAN trained without the stereochemical term in the generator loss.

reference simulations used for evaluating idpGAN (null-MD row in Table 1). This baseline essentially captures the variability of sampling between different MD simulation trajectories of the same system.

The contact mean squared error (MSE_c) quantifies differences in residue contact maps. MSE_c values from idpGAN ensembles are close to zero for almost all proteins in the IDP_test set, with the example protac discussed above (MSE_c = 2.72) being the largest outlier. In contrast, polyAla ensembles have much larger MSE_c values.

Average distances in the generated distance matrices were evaluated according to the distance mean squared error (MSE_d). Again, the idpGAN-generated average distance maps closely resemble the MD ones (Supplementary Fig. 7), and their MSE_d scores are much better than those obtained from polyAla ensembles.

We then evaluated further how well idpGAN models not just averages but distance distributions based on the average Kullback-Leibler divergence for distance distributions (aKLD_d). Again, there was close agreement between idpGAN and MD ensembles, and much better distributions can be obtained with idpGAN than from polyAla

samples. This is further illustrated by randomly selected histograms for interatomic distance data (Supplementary Fig. 8).

We continued to test whether idpGAN correctly captured not just pairwise distributions but correlations between multiple pairs. To that extent, we compared multi-dimensional joint distributions comprising all interatomic distances in proteins. To evaluate their divergence between different ensembles we used the earth mover's distance via distance root mean square deviation (EMD-dRMSD) metric, based upon an approximation[39]. As Fig. 4 shows, there is some divergence between idpGAN and MD distributions in this rather stringent metric, but the average EMD-dRMSD value of idpGAN for the IDP_test set (0.556 nm) is again lower than what is obtained with polyAla data (0.733 nm). Moreover, there is a similar degree of divergence when snapshots are taken from a separate long MD trajectory and compared with the reference MD ensemble (Supplementary Fig. 9). This suggests that the larger divergence in this metric is more likely due to incomplete sampling in the reference MD ensemble than due to poor performance of idpGAN.

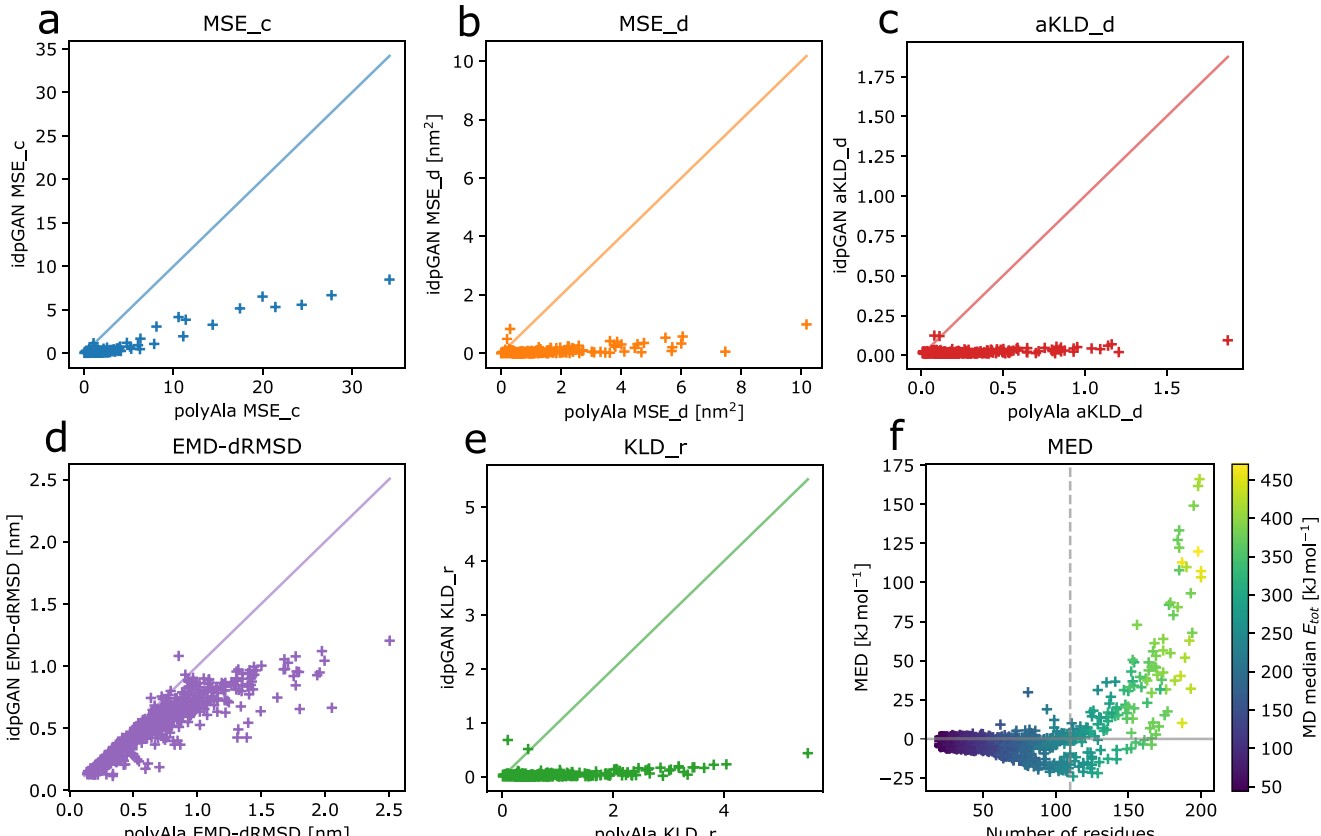

**Fig. 5 | Evaluation of idpGAN and polyAla ensembles for approximating MD data.** Results are reported for HB_val set protein ($n = 1021$). **a**, **b**, **c**, **d**, and **e** show the values of MSE_c, MSE_d, aKLD_d, EMD-dRMSD and KLD_r, respectively, obtained by polyAla (x-axis) and idpGAN (y-axis) for all the proteins in the set. See "Methods" for a description of the metrics. Lower values indicate better performance in approximating MD ensembles. IdpGAN MED values as a function of protein length are shown in **f** with markers colored according to the median potential energy of proteins in the MD ensembles. The dashed vertical line represents the maximum crop length used in idpGAN training ($L = 110$). Source data are provided as a Source Data file.

Finally, energies were compared in terms of median energy differences (MED). While the distributions do not match perfectly, there is considerable overlap and MED values are small, on average 5.6 kJ mol$^{-1}$ for the proteins in the IDP_test set. The differences increase as proteins become larger and as the average values of the energies themselves increase. We note that it was crucial to include a stereochemical term in the idpGAN generator loss function to obtain such conformations with low energies. We also trained and evaluated an idpGAN version in which this term was not used in the training objective. According to most evaluation metrics, the performance of this ablated model does not change much (Table 1), but the average MED values are much larger due to close contact clashes (Supplementary Fig. 10).

Taken together, these results show that idpGAN generates energetically-stable conformations capturing the variability of the ensembles in MD data and their amino acid sequence-specific characteristics in a transferable manner.

We further evaluated idpGAN on the larger HB_val set obtained through cross-validation from our training set. As for the IDP_test set, idpGAN provided very good approximations of the MD reference data and much better approximations with respect to random polymer ensembles from polyAla (Fig. 5 and Table 2).

Evaluating idpGAN on more IDPs permitted us to examine how its performance is affected by specific sequence features. Protein length is an important factor. Figure 5f shows that MED values of IDPs with lengths above 150 frequently surpass 50 kJ mol$^{-1}$ and the values increase for longer IDPs. Note that the maximum IDP crop length used in idpGAN training is 110. Therefore, it seems that the model has difficulties in generating energetically stable conformations for proteins

longer than the longest training protein. In Supplementary Fig. 11d, we show the distribution of each energy term of DP02478r001, the HB_val IDP with the highest MED value (166.1 kJ mol$^{-1}$). In the generated ensembles, the terms with higher median values with respect to the reference distributions are the bond length, bond angle, and short-range interactions. Despite high MED values, the distributions of the corresponding geometrical features (Supplementary Fig. 11e) and several properties of the ensemble (such as the contact map and the distribution of radius-of-gyration, Supplementary Fig. 11a to c) appear to be well-captured by the model. The higher energies result from quadratic or higher polynomial functions in the potential where even small divergences in the distributions of the features lead to large energies. Other evaluation metrics are also influenced by protein length, although for most of them the effect is weaker (Supplementary Fig. 12). This apparent limitation of idpGAN with respect to sequence length could be overcome by training with longer crops or by adding a neural-based refinement post-processing step[25].

Using the HB_val set we also evaluated whether idpGAN is overfitting on its training data. The results for the HB_val proteins were obtained with five idpGAN models and each model was evaluated on a different partition of the HB_val set, which was excluded from its training. By evaluating on the HB_val set the version of idpGAN trained with the full training set, we could determine how the model performs on proteins used in its training (see the "idpGAN all-training" row in Table 2). According to most metrics, this model is only slightly better with respect to the models that did not use the HB_val proteins (Supplementary Fig. 13). This suggests that idpGAN does not suffer from overfitting.

**Table 2 | Evaluation of idpGAN for the HB_val set**

| Method | MSE_c | MSE_d [nm²] | aKLD_d | EMD-dRMSD [nm] | KLD_r | MED [kJ mol⁻¹] |
|---|---|---|---|---|---|---|
| polyAla | 0.78 ± 0.06 | 0.32 ± 0.02 | 0.09 ± 0.004 | 0.47 ± 0.01 | 0.31 ± 0.02 | - |
| idpGAN | 0.14 ± 0.01 | 0.02 ± 0.002 | 0.02 ± 0.0002 | 0.37 ± 0.01 | 0.02 ± 0.001 | 0.83 ± 0.50 |
| idpGAN all-training[a] | 0.11 ± 0.01 | 0.01 ± 0.001 | 0.02 ± 0.0001 | 0.37 ± 0.01 | 0.02 ± 0.001 | 3.63 ± 0.47 |
| idpGAN no-stereo[b] | 0.16 ± 0.02 | 0.01 ± 0.002 | 0.01 ± 0.0002 | 0.37 ± 0.01 | 0.02 ± 0.001 | 84371 ± 15745 |

Average values are reported along with standard errors for all the proteins in the set ($n$ = 1021). Source data are provided as a Source Data file.
[a]idpGAN trained using the entire training set (which is the same model evaluated in Table 1).
[b]idpGAN trained without the stereochemical term in the generator loss.

To further assess whether idpGAN memorizes peptide geometries found in the training set, we returned to the IDP_test proteins and performed nearest neighbor searches of their generated conformations over the training set. As a control, we performed similar searches of MD conformations of the same proteins over the training set. If idpGAN were to memorize its training data, the scores of the nearest neighbor searches of its conformations would be significantly lower than the scores of the control searches conducted with MD conformations of the IDP_test proteins (which are not present in the training set). The level of divergence observed in the two searches is essentially the same (Supplementary Fig. 14), indicating that idpGAN does not simply memorize the geometrical information in the training set.

### IdpGAN based on all-atom implicit solvent simulations

We then re-trained idpGAN with data from all-atom implicit solvent Markov chain Monte Carlo (MCMC) simulations using the ABSINTH[29] potential that can accurately recapitulate experimentally-determined properties of several IDPs[40,41]. Because simulations with ABSINTH are more expensive than CG simulations, we had to limit training to peptides with less than 40 residues. Moreover, we used only the Cα atoms extracted from the all-atom trajectories to avoid the need for higher-capacity networks and accomplish training with limited resources. Even so, the high-resolution potential used in ABSINTH simulations gives rise to more complex features and the main test here is in fact whether the idpGAN model can capture such detailed features.

Re-training of idpGAN based on the ABSINTH sampling data was successful after modifying some network and training aspects (cf. "Methods"). The model was tested on 15 peptides not included in the training set (ABS_test, Supplementary Table 2), for which ABSINTH simulations were compared with experiments previously[40].

Examples of generated ensembles for ABS_test peptides (P27205, P02338.0, Q9EP54, and Q2KXY0) are shown in Fig. 6 (Supplementary Figs. 15 and 16 show other peptides). For most peptides, idpGAN correctly captured the relevant features of the reference MCMC ensembles. For example, the sequence-specific patterns in their reference free energy landscapes, contact maps, and distance maps are recovered in the ensembles generated by idpGAN for P27205, P02338.0, and Q9EP54.

To further verify that the model has learned sequence-specific conformational properties, we compared it with excluded volume simulations of the ABS_test peptides where sequence-specific interactions are turned off. IdpGAN significantly outperforms the excluded volume simulations in approximating the reference ensembles, indicating that it did not simply learn generic polymer physics (Supplementary Fig. 17 and Supplementary Table 3).

To further analyze whether idpGAN can model sequence-specific backbone structures, we compared the MCMC and generated distributions for torsion angles formed by consecutive Cα atoms in the peptides (Supplementary Fig. 18). The distributions show that idpGAN can capture the main preferences for each peptide. Moreover, we estimated the level of helicity in the ensembles (Supplementary Fig. 19). Since idpGAN produces only Cα traces, we reconstructed all-atom structures via the MMTSB Tool Set[42] before analyzing secondary

structures with DSSP[43] as implemented in MDTraj[44]. To compare against the reference distributions, we applied the same protocol to the Cα traces of the MCMC conformations of the peptides. IdpGAN generally reproduces the helical fraction in the ensembles (Supplementary Fig. 19) and approximately recovers helical frequencies at a residue-level (Supplementary Fig. 20). However, it tends to underestimate helical fractions compared to the reference ensembles, presumably because of limited training data with respect to ordered secondary structures.

For almost all peptides, idpGAN ensembles could also be used to accurately estimate properties originally calculated in Mao et al.[40], such as average radii of gyration and the scaling of internal distances along sequence separation (Supplementary Fig. 21).

Interestingly, the ABS_test set contains several peptides with high net positive charges, such as P27205 (Fig. 6) with an average charge per residue of 0.326. Although the training set that we used has an average net charge per residue of 0.018 and does not contain many highly positively charged peptides (Supplementary Fig. 22), idpGAN managed to approximately model the ensembles of these peptides, including realistic ensembles of an artificial poly-arginine sequence (Supplementary Fig. 16).

The only ABS_test peptide for which idpGAN did not fully capture the main ensemble properties is Q2KXY0 (Fig. 6). The reference ensemble of this peptide appears to have several metastable states in the PCA energy landscape and its average Cα radius-of-gyration is the lowest in the ABS_test set (Supplementary Fig. 21). This suggests that in ABSINTH simulations, this peptide behaves more like a globular peptide than an IDP. The idpGAN ensemble is a more disordered version of the reference ensemble. This may be expected since the training set focused on intrinsically-disordered regions. Another cause could be that the GAN model may not have enough capacity to model more complex multi-modal ensembles. We leave the exploration and implementation of strategies to overcome such challenges for future work.

### IdpGAN based on all-atom explicit solvent trajectories

As a final test for idpGAN, we modeled Cα traces of explicit solvent MD trajectories of a large IDP. Because explicit solvent simulations are very expensive, we focus on only one system, α-synuclein (140 residues), using trajectories from previous work[30]. Therefore, this re-trained idpGAN is an unconditional generative model that is not transferable to other sequences. To adapt idpGAN to this new data, the G network and training objective were modified. (cf. "Methods"). Supplementary Fig. 23a shows the validation of a model that was trained based on snapshots from one 2 μs all-atom simulation by comparing against MD ensembles extracted from two different, independent trajectories[30]. The contact maps show the same overall structure, but with moderate deviations from the MD-based contacts (MSE_c = 5.78) and some differences in the detailed features that indicate overfitting to features specific to the training trajectory, such as contacts between the regions around residues 60 and 130 (Supplementary Fig. 23c). The distribution of the first two principal components obtained by PCA also hints at overfitting (Supplementary Fig. 23b and **d**) as the histograms of idpGAN and training MD data are more similar than the idpGAN one

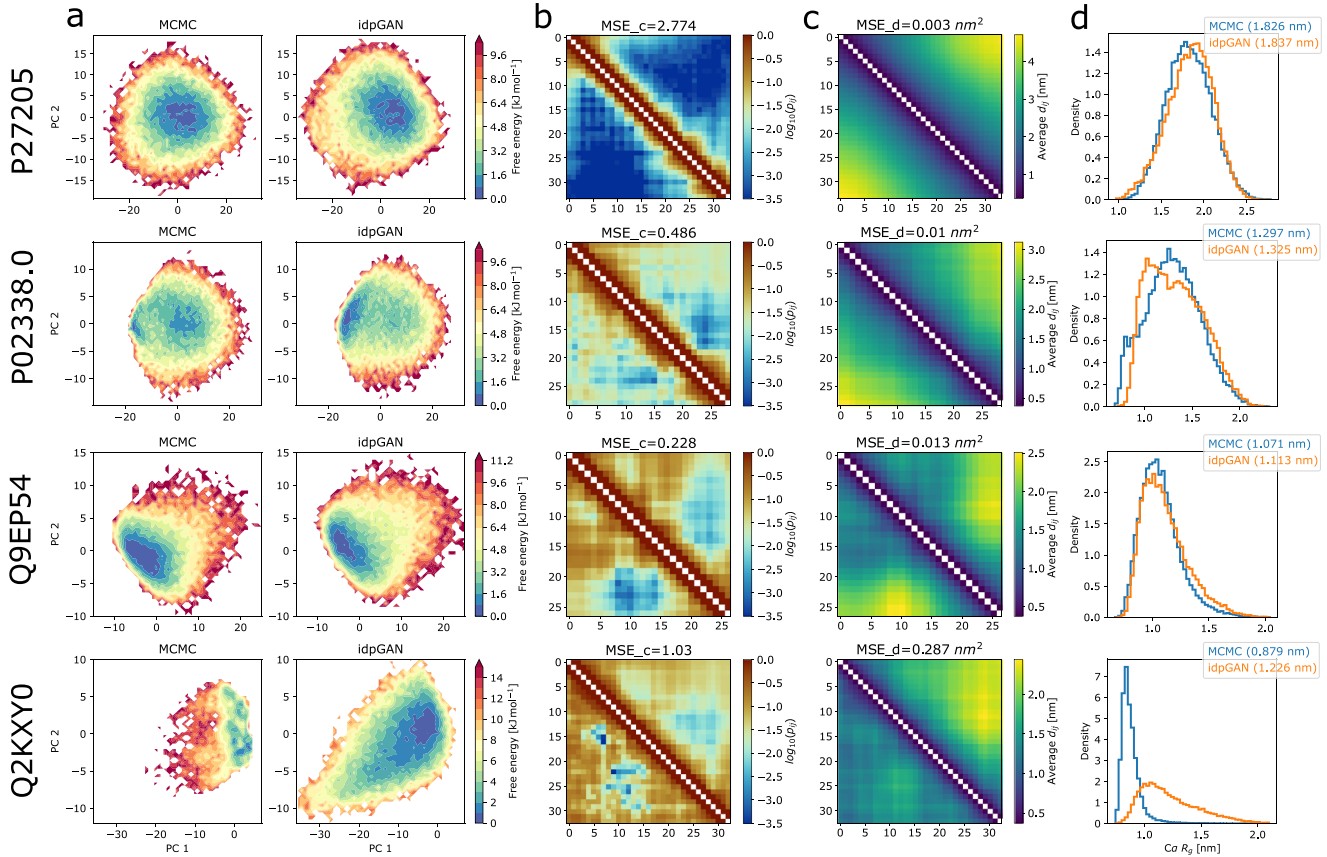

**Fig. 6 | Examples from idpGAN for four ABS_test peptides.** The rows of the image show data for P27205 ($L = 34$), P02338.0 ($L = 29$), Q9EP54 ($L = 27$) and Q2KXY0 ($L = 33$), respectively. For each peptide, the following information is reported: **a** potential of mean force profiles of MCMC (reference) and idpGAN ensembles in PCA space. For each peptide, the profiles were obtained as in Fig. 3 (see above). **b** Cα contact map evaluation, idpGAN contact maps are shown in the upper triangles of the images and MCMC maps in the lower triangles (see Fig. 2 for details). **c** Average Cα distance map evaluations, idpGAN maps are shown in the upper triangles and MCMC ones in the lower triangles. **d** Cα radius-of-gyration distributions for the MCMC (blue) and idpGAN (orange) ensembles. The average values for each distribution are shown in brackets.

compared to validation MD data. The average distance maps are similar ($MSE\_d = 0.25$ nm²) and the generated distance distributions are overall correct (Supplementary Fig. 24). The radius-of-gyration distributions are also similar ($KLD\_r = 0.30$).

These results show that a network like idpGAN has the potential to model even more complex all-atom protein conformational ensembles obtained with explicit solvent simulations. We speculate that if an idpGAN-like model can succeed in this unconditional task, it should be able to generalize to other sequences if trained with enough trajectory data from diverse proteins.

## IdpGAN sampling speed

One of the main goals of idpGAN is to obtain better computational efficiency than MD. Sampling with a GAN is very fast since it only takes a forward pass of the G network, which is highly efficient with modern deep learning libraries. For proteins with lengths below 150 residues, it typically takes less than 1 s wall-clock time to generate ensembles with thousands of independent conformations (Supplementary Fig. 25).

To compare the sampling speed of CG MD simulations and the idpGAN generator, we measured the GPU time used by both to generate enough samples to recover the distribution of the radius-of-gyration observed in 5 μs MD runs. The MD runs are already highly efficient as they involve a CG model run on a GPU. Figure 7a shows the KLD_r of idpGAN and MD ensembles containing increasing numbers of samples when compared with the long MD ensembles. For both the G network and MD simulations, KLD_r improves as more conformations are sampled and eventually reaches zero for MD. For his5, protac, and

htau23k17, the computational time it takes for the G network to reach a plateau in KLD_r (referred to as $t_{gen}$) is always less than 3 s. The time it takes for an MD simulation to reach the same KLD_r values (referred to as $t_{MD}$) is always above 250 s. IdpGAN does not perfectly recover the long MD run distributions, but it can generate close approximations based on KLD_r (Fig. 2c) orders of magnitude faster than MD. The rest of the IDP_test proteins show similar trends (see Fig. 7b). IdpGAN is more efficient relative to MD simulations for shorter proteins where the G network can generate more conformations in parallel on a GPU.

Sampling efficiency of the idpGAN model trained on ABSINTH conformational ensembles was also compared with the efficiency of MCMC simulations performed via the CAMPARI package[45]. The average $t_{MCMC}/t_{gen}$ ratio for the ABS_test peptides is $5.3 \times 10^4$ (Supplementary Fig. 26). This indicates that idpGAN can approximate the radius-of-gyration distribution of reference ensembles (Fig. 6) order of magnitudes faster than the MCMC setup we used for this benchmark.

## Discussion

The machine learning model idpGAN demonstrates the ability to directly generate realistic conformational ensembles of protein structures. This GAN-based model trained on CG or all-atom data can generate physically-realistic conformations for previously unseen proteins. The model directly generates structures that can make up a complete ensemble of energetically favorable conformations. There is no physics-based iterative sampling, which makes the approach fast. A drawback is the loss of kinetic information since generated conformations are independent from each other. However, kinetics could

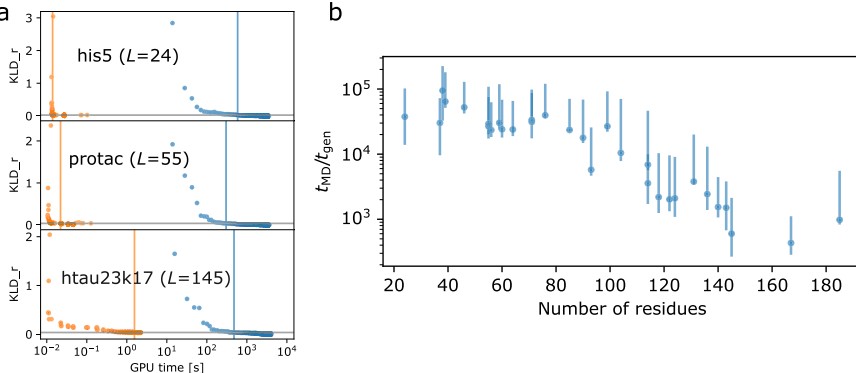

**Fig. 7 | Evaluation of idpGAN sampling efficiency compared to MD. a** KLD_r between the idpGAN and reference MD ensembles (orange data points) and between short MD and reference MD ensembles (blue data points) as a function of GPU time. Orange vertical lines indicate the GPU time ($t_{gen}$) at which the generator reaches a plateau KLD_$r_{top}$ value, which is marked by a gray horizontal line. The blue vertical lines indicate the GPU time ($t_{MD}$) needed by MD to surpass the KLD_$r_{top}$ value. Data is shown for his5, protac, and htau23k17 proteins with $t_{gen}$ values of $1.4 \times 10^{-2}$ s, $2.2 \times 10^{-2}$ s, and 1.6 s and $t_{MD}$ values of 587.7 s, 302.1 s, and 477.2 s, respectively, for the three proteins. **b** $t_{MD}/t_{gen}$ ratio for all proteins of the IDP_test set are plotted as a function of protein length. Data are presented as mean values of 10 runs (blue data points) and the bars report minimum and maximum values across the runs. Source data are provided as a Source Data file.

be recovered via simulation-based re-sampling methods[46] if the conformational ensemble is known.

Training of the model relied on conformational sampling extracted from molecular mechanics simulations. Ideally, one would like to learn from experimental data, but unfortunately, there is not much high-resolution data on conformational sampling, especially for more flexible elements in large macromolecules, such as proteins. Consequently, the machine learning model mirrors the physical realism as well as any artefacts from the simulations. We showed that a generative model can reproduce ensembles derived with different force fields and we do not see any reason why an idpGAN-like model could not learn to generate ensembles reflecting other similar energy functions or simulation protocols. One strategy in the future may be to train using simulations with multiple force fields and introduce additional constraints to incorporate experimental knowledge during the training to increase physical realism.

The simulations used for training need to be long enough to recover the underlying equilibrium distributions of conformations, as training with incomplete data could result in a generative model learning the wrong distribution. This is a difficult-to-meet requirement especially for longer sequences and more complex energy functions, although a solution for this problem might come in the form of Boltzmann generators[28].

An important point of any machine learning approach is transferability without which there is limited utility. We demonstrate that our conditional GAN model learned sufficiently general features to be able to predict correct ensembles for sequences not included in the training data. The current model works well for sequences up to and slightly beyond the longest sequence in the training set (up to 200 for the CG model and up to 40 for the ABSINTH-based model). Presumably, longer sequences could be modeled better after training with larger proteins.

In our GAN framework, the use of multiple simple neural networks as discriminators is probably inefficient and likely limits performance. We believe that employing models with stronger inductive biases as discriminators, such as graph neural networks[35], may improve our method. Additionally, our training objective uses a very simple physical-based term to improve the stereochemical quality of the generated conformations. Using more sophisticated and effective ways of including prior physical-based constraints in the learning process of the model will also be important for improving its performance. From a generative model point of view, although GANs are powerful models, their training instability[47] creates practical challenges. Therefore, other kinds of generative models may improve our

approach. Research in generative modeling is flourishing[48,49] and new methods, such as probabilistic diffusion models, are continuously being developed as possible alternatives.

For practical reasons, *i.e.* ease of generating simulation data and ease of training, our model was principally destined for CG conformations. However, we show that the approach can be extended to all-atom representations by limiting predictions to Cα traces. As the number of atoms to model increases, the training time of a generative model becomes much longer. Training a model on large-scale all-atom MD protein datasets would also require significant resources on its own. However, there is no fundamental reason why our approach could not be extended to a broader model that eventually predicts conformational ensembles of any protein at the atomistic level, given that algorithmic and hardware capabilities advance accordingly. Nevertheless, depending on the application, the current approach focusing on Cα representations of peptides of limited length may already be sufficient, for example, to generate approximate radius-of-gyration distributions of IDPs for interpretation of experimental data[30].

We demonstrate that once a generative model is trained on MD data, it can sample from the underlying distributions orders of magnitude faster than simulation-based methods. However, a central advantage of physics-based approaches, like MD, is that changes in physical conditions (temperature, pH) or chemical composition (different solvents or the presence of other solutes) are, at least in principle, easily incorporated. A machine learning model not trained with data reflecting such external variations, will not be able to provide any insights on such factors. On the other hand, generating comprehensive training data for a variety of conditions and a variety of systems, for example including complexes, is probably not practical at the current time. Therefore, future work will focus on models based on neural networks incorporating stronger inductive biases for molecular data[35] to mitigate the dependence on the amount of training data.

## Methods
### Training and test sets for the CG-based idpGAN
The training set of idpGAN consists of CG MD data for IDPs. A test set of 31 IDPs (IDP_test, Supplementary Table 1) consists of proteins with experimental radius-of-gyration values covering different sequence lengths. Some of these proteins are actual IDPs under biological conditions while others are natively folded proteins characterized in the presence of denaturant. The training set was constructed by selecting all IDPs from DisProt[50] (version 2021_06) with lengths ranging from 20 to 200 residues. Many of these IDPs are intrinsically disordered regions in larger proteins, which we neglect here. To ensure that peptides with

similar sequences are not present in both training and test sets, we removed 32 IDPs from the initial training set because of sequence similarity with proteins in IDP_test. We define a query sequence as "similar" to a training sequence based on an E-value <0.001 in a phmmer search[51] when scanning the training set with default parameters. This yielded a final training set with 1,966 IDPs (Supplementary Fig. 27).

## CG MD simulations

Conformational data for training and testing idpGAN was obtained via MD simulations for all the training and test set IDPs using a recently developed CG model from our group[52]. In this model, each residue is represented as a single spherical particle located at the Cα atom of a given residue. The potential form is given by:

$$U_{\text{total}} = \sum_{i=1}^{L-1} \frac{1}{2} k_{\text{bond}} \left( l_{i,i+1} - l_0 \right)^2$$
$$+ \sum_{i=1}^{L-2} \frac{1}{2} k_{\text{angle}} \left( \theta_{i,i+1,i+2} - \theta_0 \right)^2$$
$$+ \sum_{i,j} 4 \left( \varepsilon + \varepsilon_{\text{cation}-\pi} \right) \left( \left( \frac{\sigma_{i,j}}{r_{i,j}} \right)^{10} - \left( \frac{\sigma_{i,j}}{r_{i,j}} \right)^5 \right) \tag{1}$$
$$+ \sum_{i,j} \frac{\left( A_{i,j} + A0_{i,j} \right)}{r_{i,j}} e^{-\frac{r_{i,j}}{k}},$$

where $L$ is the number of residues in a protein. The bonded parameters are as follows: $l_{i,i+1}$ is the distance between two neighboring residues, with the spring constant $k_{\text{bond}} = 4{,}184$ kJ (mol·nm²)⁻¹, $l_0 = 0.38$ nm is the equilibrium bond length; $\theta_{i,i+1,i+2}$ is the angle between two subsequent Cα beads, with an angle spring constant of $k_{\text{angle}} = 4.184$ kJ (mol·rad²)⁻¹, and an equilibrium angle $\theta_0 = 180^o$. The remaining terms refer to non-bonded interactions: $r_{i,j}$ is the inter-residue distance for residues not connected via bonds, $\sigma_{i,j} = \sigma_i + \sigma_j$ where $\sigma_i$ was determined as the radius of a sphere with an equivalent volume of a given residue, $\varepsilon$ is set to 0.40 and 0.41 kJ mol⁻¹ for polar and non-polar residues, respectively, $\varepsilon_{\text{cation}-\pi}$ is set to 0.3 kJ mol⁻¹ to augment interactions between basic residues (Arg/Lys) and aromatic residues (Phe/Tyr/Trp); $A_{i,j} = A_i \times A_j$ describes long-range interactions with $A_i = \text{sign}(q_i)\sqrt{0.75|q_i|}$[53] using charges $q_i = +1$ for Arg/Lys, $q_i = -1$ for Asp/Glu, and $q_i = 0$ for all other residues; $A0_{i,j} = A0_i \times A0_j$ describes the repulsion between polar residues due to solvation with $A0_i$ being 0.05 for polar and 0 for non-polar residues, respectively.

For all IDPs as well as the polyAla reference, simulations were run with OpenMM 7.7.0[54] via Langevin dynamics with a friction coefficient of 0.01 ps⁻¹. An initial equilibration was performed with 5000 steepest descent minimization steps followed by 20,000 steps of MD with a 0.01 ps time step. For production runs we used a time step of 0.02 ps. Periodic boundary conditions were applied and non-bonded interactions were truncated at 3 nm. Bonded residues were excluded in non-bonded interaction evaluations. Individual protein chains were simulated in a cubic box of side 300 nm at 298 K. For all proteins we ran five separate trajectories over 1000 ns and one additional longer trajectory over 5000 ns for the proteins in the IDP_test set (corresponding to the null-MD row in Table 1). Coordinates were saved every 200 ps giving a total of 25,000 trajectory snapshots for each protein. Initial random coordinates for each chain were obtained using a custom Python script. Topology files were generated via the MMTSB Tool Set[42] and CHARMM v44b2[55].

## Training objective for CG-based idpGAN

The learning process of a GAN involves the training of two neural networks, G and D[31,56]. IdpGAN is a conditional GAN[57], trained with the non-saturated GAN objective[31,58]. The discriminator loss, denoted as $L_D$, is:

$$L_D = -\mathbb{E}_{\mathbf{x} \sim p_d, \mathbf{a} \sim p_a} \left[ \log(D(\mathbf{x}, \mathbf{a})) \right] - \mathbb{E}_{\mathbf{z} \sim p_z, \mathbf{a} \sim p_a} \left[ \log(1 - D(G(\mathbf{z}, \mathbf{a}), \mathbf{a})) \right], \tag{2}$$

where $p_d$ is the distribution in the training set of examples $\mathbf{x}$ (describing molecular conformations), $p_a$ is the distribution of examples $\mathbf{a}$ (representing amino acid sequences), and $p_z$ is the prior from which $\mathbf{z}$ values are sampled. The discriminator output $D(\mathbf{x}, \mathbf{a})$ is a scalar from 0 to 1 and represents the probability of a sample $\mathbf{x}$ with sequence $\mathbf{a}$ to be real. The output of $G(\mathbf{z}, \mathbf{a})$ is a generated conformation $\mathbf{x}$ for a protein with sequence $\mathbf{a}$.

For the generator loss, we added a term $E_C$ to the original non-saturated GAN loss, inspired by a term in the AF2 objective[10], to reduce steric clashes between non-bonded atoms (atoms in residues three or more positions apart). The term takes as input a vector $\mathbf{x}$ (which stores all interatomic distances in a conformation, see below) and is expressed as:

$$E_C(\mathbf{x}) = \sum_{j \in N(\mathbf{x})} \max\left( x_t - x_j, 0 \right), \tag{3}$$

where $N(\mathbf{x})$ is the set of indices for all non-bonded distances in $\mathbf{x}$ and $x_t = 0.59$ nm is a threshold corresponding to the 0.1 percentile of the training set non-bonded distances. The generator loss, denoted as $\hat{L}_G$, is therefore:

$$\hat{L}_G = \mathbb{E}_{\mathbf{z} \sim p_z, \mathbf{a} \sim p_a} \left[ -\log(D(G(\mathbf{z}, \mathbf{a}), \mathbf{a})) + w_c E_c(G(\mathbf{z}, \mathbf{a})) \right], \tag{4}$$

where $w_C = 0.3$.

## Generator and discriminator networks for CG-based idpGAN

The idpGAN neural networks were implemented using the PyTorch framework[59]. The G network has a transformer architecture[34] (Supplementary Fig. 1). For a protein of length $L$, the network takes as input: a tensor $\mathbf{z} \in \mathbb{R}^{L \times n_z}$ with values randomly sampled from a Gaussian $N(0,\mathbf{I})$ and a tensor $\mathbf{a} \in \mathbb{R}^{L \times 20}$ with one-hot encodings for the amino acid sequence of the protein. The output of the network is a tensor $\mathbf{r} \in \mathbb{R}^{L \times 3}$, representing the 3D coordinates of the protein Cα atoms.

Four MLP discriminators were employed. Each discriminator has the same architecture and hyper-parameters, with a non-trainable standardization layer, three linear layers with spectral normalization[47] to regularize GAN training, two leaky ReLU non-linearities and a final sigmoid activation (Supplementary Table 4). The input of a discriminator is composed as follows: starting from a conformation $\mathbf{r}$, we compute its distance matrix $\mathbf{X} \in \mathbb{R}^{L \times L}$, extract its upper triangle (excluding its zero-filled diagonal) and flatten it to a vector $\mathbf{x} \in \mathbb{R}^{L(L-1)/2}$. We then process $\mathbf{x}$ through a standardization layer where each value $x_i$ (representing a distance between two Cα atoms with sequence separation $k$) is converted via $\hat{x}_i = (x_i - m_k)/s_k$, where $m_k$ and $s_k$ are the mean and standard deviation in the training set for distances between Cα atoms with separation $k$. To provide amino acid sequence information, $\hat{\mathbf{x}}$ is concatenated to a flattened version of $\mathbf{a}$ (a 20$L$-dimensional vector). The resulting $L(L-1)/2 + 20L$-dimensional vector is the input of the first linear layer of the MLP.

## Training process for CG-based idpGAN

A set of four length values $L_{\text{train}} = (20,50,80,110)$ was chosen. For each value, we employed a MLP discriminator that only takes as input proteins crops with that length. To make use of all IDPs, at the beginning of each training epoch we adopt the following strategy: (1) Each IDP with a length value present in $L_{\text{train}}$ is associated with the corresponding MLP. (2) For IDPs with length values not in $L_{\text{train}}$, we crop them to the closest value in $L_{\text{train}}$. In this way, 1,070 IDPs were assigned to $L = 20$, 317 to $L = 50$, 246 to $L = 80$, and 279 to $L = 110$. (3) To

avoid unbalanced training, we associate $c_{max} = 1{,}070$ IDPs to all $L_{train}$ values by randomly sampling all training IDPs and assign each to a random $L_{train}$ value, until all MLPs are associated with $c_{max}$ IDPs. (4) Each time an IDP is assigned to a MLP, we randomly sample $n_{frames} = 1{,}750$ frames from its MD data. When an IDP is cropped, each frame is randomly cropped selecting different starting and ending residues. This random selection scheme at each epoch is effectively a form of data-augmentation. By using this strategy, we have $|L_{train}| \times c_{max} \times n_{frames} = 4 \times 1{,}070 \times 1750 = 7{,}490{,}00$ training MD frames per epoch.

To optimize the idpGAN objective, we use Adam optimizers[60] (with $\beta_1 = 0.0$ and $\beta_2 = 0.9$ hyper-parameters) and employ learning rates of 0.00025 and 0.0004 for G and all D networks (each D network has its own optimizer).

For training, we use a batch size of 192, with batches containing crops having the same number of residues. The training of an idpGAN model lasts for 50 epochs (roughly three days on an NVIDIA RTX2080Ti GPU). The training was repeated ten times and the model with the best performance on validation data was selected.

### Evaluation strategy for CG-based idpGAN

For initial evaluation of idpGAN, we used the IDP_test set. To evaluate idpGAN on more proteins, we also split the training set into partitions based on sequence similarity[61]. By performing an all-to-all search with phmmer, we identified 1,021 IDPs that do not have other similar sequences in the set. Removing this subset, which we call the HB_val set, the remaining training set would likely be too small to obtain good performance in terms of generalization to arbitrary sequences. Therefore, we randomly split the HB_val subset into five approximately equally-sized partitions. For each partition, we trained idpGAN with all remaining IDPs, including the four other partitions of the HB_val set, and then validated it on the IDPs of the selected partition itself. By repeating this procedure with all partitions, we were able to evaluate the performance of idpGAN on the HB_val set.

### Evaluation metrics

To compare idpGAN (or polyAla) ensembles with the reference (MD) ones, we used different metrics based on ensembles with 10,000 randomly sampled conformations.

Contact probabilities for a protein of length $L$ were evaluated with:

$$\text{MSE\_c} = \frac{1}{N_{pairs}} \sum_{i<j} (\log(p_{ij}) - \log(\hat{p}_{ij}))^2, \quad (5)$$

where $N_{pairs} = L(L-1)/2$ is the number of residue pairs in the protein, $p_{ij}$ and $\hat{p}_{ij}$ are the contact frequencies for residues $i$ and $j$ in the reference and generated ensembles respectively (a pseudo-count value of 0.01 was used for zero frequencies). A C$\alpha$ distance threshold of 8.0 Å was used to define a contact.

Average interatomic distance values were evaluated via with:

$$\text{MSE\_d} = \frac{1}{N_{pairs}} \sum_{i<j} (m_{ij} - \hat{m}_{ij})^2, \quad (6)$$

where $m_{ij}$ and $\hat{m}_{ij}$ are the average distance values between the C$\alpha$ atoms of residue pair $i$ and $j$ in the reference and generated ensembles. Monodimensional distributions of continuous features were compared by an approximation of the Kullback-Leibler divergence (KLD): the range between minimum and maximum values of a feature over the reference and generated ensembles was divided into $N_{bins} = 50$ bins. From the frequencies of observing values in each bin (with a pseudo-count value of 0.001) an approximate KLD was computed as:

$$\text{KLD}(P \parallel Q) = \sum_k^{N_{bins}} P_k \log\left(\frac{P_k}{Q_k}\right), \quad (7)$$

where $k$ is the index of a bin and $P_k$ and $Q_k$ are the frequencies associated with bin $k$ for the reference and generated ensembles respectively.

Pairwise interatomic distance distributions were compared according to:

$$\text{aKLD\_d} = \frac{1}{N_{pairs}} \sum_{i<j} \text{KLD}(M_{ij} \parallel \hat{M}_{ij}), \quad (8)$$

where $M_{ij}$ and $\hat{M}_{ij}$ are the distance distributions between the C$\alpha$ atom of residue $i$ and $j$ in the reference and generated ensembles.

Multi-dimensional joint distributions of all interatomic distances in proteins were compared by approximating their earth mover's distance (EMD) based on the distance root mean square deviation (dRMSD) between conformations[39]. Given a generated and reference ensemble, we compute dRMSD values for all pairs of conformations. The dRMSD for two conformations is:

$$\text{dRMSD} = \sqrt{\frac{1}{N_{pairs}} \sum_{i<j} (d_{ij} - \hat{d}_{ij})^2}, \quad (9)$$

where $d_{ij}$ and $\hat{d}_{ij}$ are the distances between C$\alpha$ of residues $i$ and $j$ in the reference and generated structures. We use the Hungarian algorithm to pair each reference conformation to a generated one, minimizing the global dRMSD. The total dRMSD is finally averaged over pairs and reported as the EMD-dRMSD metric.

Radius-of-gyration distributions were compared according to:

$$\text{KLD\_r} = \text{KLD}(R \parallel \hat{R}), \quad (10)$$

where $R$ and $\hat{R}$ are the distributions of radius-of-gyration for the reference and generated ensembles.

### Modeling C$\alpha$ traces from all-atom implicit solvent simulations

Training data for modeling C$\alpha$ traces of all-atom peptides was collected via Metropolis MCMC sampling with the OPLS-AA/L force field[62] and the ABSINTH implicit solvation model[29] using CAMPARI 4.0[45]. We simulated all 919 peptides with 40 or less residues in the idpGAN training set. For each peptide, we performed five independent simulations following a previously described CAMPARI protocol[40] (Supplementary Note 1).

The same protocol was employed to run 20 simulations for the 15 (of the 21) peptides presented in Mao et al.[40] having 40 or less residues. These 15 sequences constitute the ABS_test test set (Supplementary Table 2). Simulation results for these peptides from Mao et al.[40] were found to be in good agreement with experimental data. Therefore, the ABS_test set allowed us to test if idpGAN can be used to obtain biophysically-relevant information. None of the ABS_test sequences have a similar sequence in the training set. For the ABS_test peptides we also ran 20 excluded volume simulations[40] as a control. In these simulations, only the repulsive part of the Lennard-Jones potential is used for non-bonded interactions.

To adapt idpGAN to the ABSINTH simulation data, the neural network and training protocol were modified. To avoid mode-collapse[56], the hinge loss[47] was used instead of the non-saturated GAN loss. Moreover, training was stabilized with the top-k training technique[63] using parameters $\gamma = 0.99$ and $\nu = 0.5$. To aid training dynamics, a Gaussian noise with $\sigma = 0.025$ nm was added to the Cartesian coordinates of the training data during the first epochs and gradually annealed throughout training.

 

We used the same architecture and hyper-parameters for the G network for the CG-based idpGAN. Only the default "post" configuration for layer normalization in the transformer blocks was changed to the "pre" one[64]. For the D network, we used a single discriminator with a 2D convolutional residual network architecture taking as input the distance matrix of conformations (Supplementary Fig. 28).

To train the new model, 1,750 conformations were randomly selected for each peptide during each training epoch. We employed the same optimizers described for the CG conformation learning task. A batch size of 192 was used with batches containing conformations having the same number of residues. We trained idpGAN for 200 epochs and decreased the learning rates of both G and D by a factor of 0.5 at epochs 50, 100, and 150. We repeated training 10 times and selected the model with the best performance on validation data.

Given that idpGAN is reflection invariant but Cα traces from all-atom proteins have a chiral probability distribution[65] (Supplementary Fig. 29), a post-processing step was added. The correct mirror image for each conformation was determined via an additional selector neural network. If a conformation was predicted to have the "wrong" handedness, it was reflected. The selector network takes as input the values of the torsion angles formed by four consecutive Cα atoms of a peptide, whose joint distribution is sufficient to discern correct handedness with a good accuracy[65]. Adding the values of these angles as features for the D network would have been an alternative solution, but we found that training was not stable by applying it to this task. The selector network has the same architecture of the G network and was trained using snapshots from ABSINTH simulations. Details are given in Supplementary Note 2.

### Modeling Cα traces from all-atom explicit solvent simulations

The training data we used for all-atom α-synuclein modeling is a 2 μs simulation containing 10,000 snapshots. For testing we compared with data from two additional independent simulations of the same system. Details of the simulations are given elsewhere[30]. To evaluate idpGAN, we generated 5,000 samples.

In this learning task, the Wasserstein loss with gradient penalty[66] produced the best results with five critic iterations and $\lambda = 10$.

To adapt idpGAN for this more complex data, we also had to modify its G network by increasing its capacity and introducing other small modifications (Supplementary Table 5). α-synuclein has $L = 140$ residues. The input size of the discriminator is constant and therefore we employed a single MLP. Its architecture is similar to the one used for modeling CG conformations (Supplementary Table 4). The discriminator input is a vector of dimension $L(L-1)/2 + (L-3)$. The $L(L-1)/2$ features account for the distances between all Cα pairs, while the $L-3$ features account for the dihedral angles between all groups of four consecutive Cα atoms. We included these latter features to force G to learn the correct mirror images for α-synuclein Cα traces. While training, we adopt a per-feature non-trainable normalization layer, as described for the CG idpGAN version. No conditional amino acid information was inputted since we considered only one system.

We employed the same optimizers described for the CG modeling task. We trained idpGAN for 300 epochs using a batch size of 64 and decreased by a factor of 0.5 both G and D learning rates at epochs 100 and 200. We repeated training 10 times and selected the model with the best performance on validation data.

### Evaluating sampling efficiency

Sampling efficiency of idpGAN for IDP_test proteins was determined as follows: (1) a 5,000 ns MD simulation was run for a given protein yielding a conformational ensemble $E_{MD}$ with 25,000 snapshots. (2) The G network was used to sample an increasing number of conformations $n$ (from 50 to 15,000). For each generated ensemble $E_{gen,n}$, KLD_r was calculated with respect to $E_{MD}$. The KLD_r values decrease as the number of samples increases, and soon reach a plateau (Fig. 7).

When KLD_r does not decrease anymore even after generating an additional 1000 samples, we stop. The minimum KDL_r value obtained in this way is KLD_r$_{top}$ and the time it took for G to generate the corresponding ensemble is $t_{gen}$. (3) We extract an increasing number of snapshots $m$ from the beginning of the long MD simulation (from frame 50 to 25,000). For each MD ensemble $E_{MD,m}$, we measure its KLD_r with $E_{MD}$. When we find an ensemble that improves over KLD_r$_{top}$, we denote as $t_{MD}$ the time taken for the simulation to accumulate the number of samples to match that performance.

All runs were performed on NVIDIA RTX2080Ti GPUs. When generating samples with idpGAN, mini-batches as large as possible were used to maximize GPU performance. Different batch sizes were used for different $L$ values to fit batches onto memory (Supplementary Table 6).

The sampling efficiency of idpGAN based on ABSINTH simulations for the ABS_test peptides was determined similarly. We run on a GPU the re-trained version of idpGAN and the mirror image selector network. The sampling speed of the neural networks is compared to the speed of 20 parallel MCMC simulations running on 4 cores each on Intel Xeon Silver 4214 CPUs (2.20 GHz). CAMPARI only runs on CPUs.

### Reporting summary

Further information on research design is available in the Nature Portfolio Reporting Summary linked to this article.

## Data availability

The training set sequences were obtained from DisProt[50] version 2021_06 available at https://disprot.org/download. Training set and IDP_test, ABS_test sequences, and HB_val validation splits are available at https://github.com/feiglab/idpgan[67]. Raw data is presented in figures and tables in the main text and Supplementary Information are provided in the Source Data file. Source data are provided with this paper.

## Code availability

The code for the idpGAN generator and its weights are available at https://github.com/feiglab/idpgan[67]. Jupyter notebooks illustrate how to use idpGAN to generate and analyze 3D conformational ensembles.

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

## Acknowledgements

This research was supported by National Institutes of Health Grant R35 GM126948 (to MF). High-performance computing resources were provided in part by the Institute for Cyber-Enabled Research at Michigan State University.

## Author contributions

G.J., L.H., and M.F. designed the research, G.J. performed and analyzed the machine learning work, G.V.G. and G.J. performed and analyzed the simulations used for training, and all authors jointly discussed the findings and wrote the manuscript.

## Competing interests

The authors declare no competing interests.
