## [Peer review file · Nature Communications]

REVIEWER COMMENTS

Reviewer #1 (Remarks to the Author):

The manuscript describes a very interesting transformer-based generative adversarial network for direct generation of the conformational ensemble of intrinsically disordered proteins (IDPs). The current state of the art approaches for generating IDP conformational ensembles are generally based on molecular dynamics or Monte Carlo simulations, which can be computationally demanding. An ability to directly generate IDP ensemble with drastically reduced computational cost would be a very important advance and have large impacts. This work is thus a timely contribution. However, I have reservations that the model as described has demonstrated an ability to generate nontrivial properties of IDPs that would make it an impactful tool.

1) The overall idpGAN network design is a standard one. The innovation of this work thus falls on how the model is trained and the level of performance achieved. As explained below, the model does not seem to have demonstrated a level of performance that would make it truly innovative.

2) The model is primarily trained using MD trajectories generated using a residue-level CG protein model that was designed to capture simple polymer properties such as radius of gyration distribution of IDPs. It should be noted that many similar CA-only models have been described for similar purposes and the computational cost of these models is quite modest. The key limitation of these models, however, is that they can not describe nontrivial but important properties of IDPs such as transient/local structures. Being able to reproduce the simple polymer properties (e.g., Fig 2) using idpGAN is nice, but does not seem surprising. I think it is necessary to go beyond the residue-level resolution to provide a high-quality description of IDP ensembles in general, especially those that do not behave like some type of (classical) polymers.

3) The authors demonstrate the potential of idpGAN to generate atomistic ensembles using a MD trajectory of a single IDP. However, this seems to be largely a glorified exercise of dimension reduction fitting of a high dimensional data. It does not provide any support for the ability to "predict sequence-dependent ensembles for any sequence".

4) I suggest that the authors consider demonstrating the potential of idpGAN using some atomistic ensembles, such as those generated by computationally efficient simple implicit solvent models. Along this line, I think the ABSINTH model developed by Rohit Pappu may provide a very good approach for generating a large number of ensembles very efficiently. This model has been shown to be very capable of describing not only the polymer properties but also some nontrivial peptide properties (such as residual

helices). A generative model that could somewhat reproduce ABSINTH ensembles would be truly remarkable and represent a significant breakthrough.

5) I think it is misleading and unnecessary to show pairs of single structures from MD and idpGAN in Fig 2 and 6. Instead, we should focus exclusively on ensemble properties, conformational distributions, major states, statistical characteristics etc.

Reviewer #2 (Remarks to the Author):

In this work, the authors used a Deep Neural Network trained on data from MD simulations so that the trained DNN can generate physically realistic conformational ensembles of proteins. The authors claimed that the developed idpGAN is efficient and requires negligible computational cost.

The work is interesting and has merits. Nonetheless, the following concerns should be addressed by the authors before the manuscript can be found acceptable for publication.

[1] Some relevant works were not acknowledged and cited. For example, <https://doi.org/10.3389/fmolb.2021.587151>; <https://doi.org/10.1109/SSCI47803.2020.9308559>; <https://doi.org/10.1021/acs.jcim.7b00690>; <https://doi.org/10.48550/arXiv.1705.10843>; and <https://doi.org/10.48550/arXiv.1805.11973>

[2] The comparison of idpGAN to other methods is inadequate and lacks holistic structural information. For example, can the authors compare the results from their method/idpGAN and the results from MD simulations using PCA-based reaction coordinates (RCs) and generate PMF/energy landscape based on PCA-based RCs or any other combination of RCs (e.g., RMSD and Radius of Gyration) that make use of all the residues in the system? For example, the last figure (Fig 8) in each of these, perhaps too simple, papers <https://doi.org/10.3389/fmolb.2021.587151> <https://doi.org/10.1109/SSCI47803.2020.9308559> used RCs like these.

[3] Where is the comparison of the MD simulation data that the DNNs were trained with (i.e., the training set) and the data that the DNN generated (i.e., the output)? Can the authors prove that the DNN did not just memorize the training data? Many readers will be interested in this.

[4] Did the authors use the model trained on one molecule to sample the conformation space of another molecule? If not, why not? If yes, where is this information in the manuscript? There should be a vivid way of conveying this to the readers. I understand the authors hinted at this in the introduction, but I do not know if this is clearly shown in specific display items.

[5] What is the largest protein that the model was tested on? Did the authors test the approach on larger and more realistic systems (such as proteins and complexes with 200 to 500 residues) that are often the subject of most MD simulations?

[6] How did the authors conclude that idpGAN is efficient and requires negligible computational cost? Vividly address this in the manuscript if not already done, and make it more obvious.

[7] How did the authors obtain the parameters for the CG models? Although the authors listed the parameter in the “Coarse-grained molecular dynamics simulations” section, it is unclear how you obtained the parameters. For example, how did the authors obtain the value for the spring constant, the equilibrium bond length, etc.? These should be clearly stated, so it is effortless for the readers to find.

[8] Did the authors use idpGAN for protein complexes (i.e., for protein systems containing more than one chain of protein)? If not, why not?

We thank the reviewers for their thoughtful comments, and we have revised the manuscript accordingly. A detailed response to the issues raised by the reviewers is given below:

Reviewer #1 (Remarks to the Author):

The manuscript describes a very interesting transformer-based generative adversarial network for direct generation of the conformational ensemble of intrinsically disordered proteins (IDPs). The current state of the art approaches for generating IDP conformational ensembles are generally based on molecular dynamics or Monte Carlo simulations, which can be computationally demanding. An ability to directly generate IDP ensemble with drastically reduced computational cost would be a very important advance and have large impacts. This work is thus a timely contribution. However, I have reservations that the model as described has demonstrated an ability to generate nontrivial properties of IDPs that would make it an impactful tool.

We thank the reviewer for the constructive and insightful comments. As a result, we believe to have substantially improved the quality of our manuscript.

1) The overall idpGAN network design is a standard one. The innovation of this work thus falls on how the model is trained and the level of performance achieved. As explained below, the model does not seem to have demonstrated a level of performance that would make it truly innovative.

We expanded our efforts by re-training idpGAN based on ABSINTH simulation data (see below). We believe that the new results demonstrate clearly that our method is able to reproduce more complex dynamic ensembles obtained with non-trivial models. Thus, we think that our efforts are truly innovative with respect to existing approaches described in peer-reviewed works of similar scope.

2) The model is primarily trained using MD trajectories generated using a residue-level CG protein model that was designed to capture simply polymer properties such as radius of gyration distribution of IDPs. It should be noted that many similar CA-only models have been described for similar purposes and the computational cost of these models is quite modest. The key limitation of these models, however, is that they can not describe nontrivial but important properties of IDPs such as transient/local structures. Being able to reproduce the basis polymer properties (e.g., Fig 2) using idpGAN is nice, but does not seem surprising. I think it is necessary to go beyond the residue-level resolution to provide a high-quality description of IDP ensembles in general, especially those do not behave like some type of (classical) polymers.

We believe that our underlying IDP model is not just capturing 'trivial' polymer behavior (see bioRxiv preprint: <https://doi.org/10.1101/2022.08.19.504518>). However, the point is well-taken. In response to point 4 below, we have re-trained idpGAN based on ABSINTH simulation data to demonstrate that our approach can reproduce more complex features of IDPs.

3) The authors demonstrate the potential of idpGAN to generate atomistic ensembles using a MD trajectory of a single IDP. However, this seems to be largely a glorified exercise of dimension reduction fitting of a high dimensional data. It does not provide any support for the ability to "predict sequence-dependent ensembles for any sequence".

We show now in the revision that the idpGAN approach can be extended to conformational ensembles with more complex features based on implicit-solvent ABSINTH training data. The consideration of even more complex features from all-atom explicit solvent simulations of IDPs is the next step. Based on the alpha-synuclein example, we demonstrate that the

idpGAN architecture is in principle also able to capture the features of such ensembles. We believe that this is a useful exercise to demonstrate the ultimate promise of idpGAN of replacing MD-based ensemble generation. We agree, however, that the all-atom/explicit-solvent based model has limited practical use because the training data is limited to just one system.

In order to better present the role of the α -synuclein modeling results in the manuscript, we modified the text at the beginning of the “***IdpGAN based on all-atom explicit solvent trajectories***” section by explaining these concepts and we highlighted the more preliminary nature of the results. Given the secondary importance of these results, we moved **Figure 6** from the original manuscript in the supplementary materials (now it is **Supplementary Figure 22**).

To make the scope of our work clearer to the reader, we also modified the sentence quoted by the reviewer (“predict sequence-dependent ensembles for any sequence”) in the **Abstract**.

4) I suggest that the authors consider demonstrating the potential of idpGAN using some atomistic ensembles, such as those generated by computationally efficient simple implicit solvent models. Along this line, I think the ABSINTH model developed by Rohit Pappu may provide a very good approach for generating a large number of ensembles very efficiently. This model has been shown to very capable of describing not only the polymer properties but also some nontrivial peptide properties (such as residual helices). A generative model that could somewhat reproduce ABSINTH ensembles would be truly remarkable and represent a significant breakthrough.

We followed the advice, adapted idpGAN and re-trained it on extensive ABSINTH simulation data that we collected. Given the computational restraints associated with running all-atom simulations on CPUs (the ABSINTH model is implemented in CAMPARI, a CPU-only package) and the time needed to fine-tuning GAN models with new datasets, we had to restrict our experiments to peptides with a maximum length of 40 residues in order to be able to revise our manuscript in a timely manner. We believe that the newly trained idpGAN can model with good accuracy the conformational ensembles of peptides simulated with the ABSINTH potential. We tested it on peptides that are not present in the training set and found that for almost all test sequences it can recapitulate the key features of the conformational ensembles. This clearly demonstrates that idpGAN can capture nontrivial polypeptide properties such as the formation of alpha-helical backbone geometries. To our knowledge, this is the first time that a conditional generative model is used to reproduce the structural ensembles from all-atom protein simulations.

The results are presented in the newly introduced “***IdpGAN based on all-atom implicit solvent simulations***” section and in **Figure 6** of the revised manuscript.

We also added several supplementary material items to account for additional results and the methodological details of the new experiments (**Supplementary Figures 15-21, 25, 27-28; Supplementary Tables 2-3; Supplementary Texts 1-2**).

5) I think it is misleading and unnecessary to show pairs of single structures from MD and idpGAN in Fig 2 and 6. Instead, we should focus exclusively on ensemble properties, conformational distributions, major states, statistical characteristics etc.

We agree that portraying just a pair of structures for each protein system can be misleading to the reader. We therefore removed them from **Figure 2** and **Supplementary Figure 22** (which now contains the material of **Figure 6** in the original manuscript).

However, we still believe that showing generated 3D structures has some qualitative value, because it visually conveys the degree of physical realism of the generated structures. For this reason, we added a supplementary item (**Supplementary Figure 6**) which shows 6 pairs of reference-generated conformations for the CG protein systems in **Figure 2**. Showing multiple examples of randomly selected reference-generated pairs is a standard practice in generative modeling literature (for example, see Figure 2 of <https://openreview.net/forum?id=PzcvxEMzvQC>) and we decided to follow it. In the rest of the manuscript, we then focus all our attention to CG ensembles properties at the structural and energetic level. In these revisions, we also extended this kind of analyses by building PCA potential of mean force landscapes of the ensembles (see point 2 of Reviewer 2 for a full discussion).

Reviewer #2 (Remarks to the Author):

In this work, the authors used a Deep Neural Network trained on data from MD simulations so that the trained DNN can generate physically realistic conformational ensembles of proteins. The authors claimed that the developed idpGAN is efficient and requires negligible computational cost.

The work is interesting and has merits. Nonetheless, the following concerns should be addressed by the authors before the manuscript can be found acceptable for publication.

We thank the reviewer for the positive and insightful comments and suggestions. By addressing all the points, we believe that we have made our work clearer and improved its quality.

[1] Some relevant works were not acknowledged and cited. For example, <https://doi.org/10.3389/fmolb.2021.587151>; <https://doi.org/10.3389/fmolb.2021.587151>; <https://doi.org/10.1109/SSCI47803.2020.9308559>; <https://doi.org/10.1109/SSCI47803.2020.9308559>; <https://doi.org/10.1021/acs.jcim.7b00690>; <https://doi.org/10.1021/acs.jcim.7b00690>; <https://doi.org/10.48550/arXiv.1705.10843>; <https://doi.org/10.48550/arXiv.1705.10843>; and <https://doi.org/10.48550/arXiv.1805.11973> <https://doi.org/10.48550/arXiv.1805.11973>

Thank you. We added the suggested references for the machine learning-based enhanced sampling methods and for the previous use of GAN models in molecular sciences.

[2] The comparison of idpGAN to other methods is inadequate and lacks holistic structural information. For example, can the authors compare the results from their method/idpGAN and the results from MD simulations using PCA-based reaction coordinates (RCs) and generate PMF/energy landscape based on PCA-based RCs or any other combination of RCs (e.g., RMSD and Radius of Gyration) that make use of all the residues in the system? For example, the last figure (Fig 8) in each of these, perhaps too simple, papers <https://doi.org/10.3389/fmolb.2021.587151> <https://doi.org/10.3389/fmolb.2021.587151> <https://doi.org/10.1109/SSCI47803.2020.9308559> <https://doi.org/10.1109/SSCI47803.2020.9308559> used RCs like these.

We followed the reviewer's suggestion and constructed for our systems a series of potential of mean force profiles or histograms (similar to those in <https://doi.org/10.1109/SSCI47803.2020.9308559>).

We added such profiles: (1) for all the test set proteins used for the modeling of the C α -based MD ensembles in **Figure 3** and **Supplementary Figures 2-5** (and we also add a paragraph to the “***IdpGAN based on CG simulations***” section to discuss these analyses); (2) for all the test set proteins used for modeling the ABSINTH simulation ensembles (see point 4 of reviewer 1) in **Figure 7**, **Supplementary Figures 15-16** and **19**; (3) for the alpha-synuclein MD ensembles in **Supplementary 22**.

Given the results presented in these figures and the analyses performed with the evaluation scores that we originally used (several of these scores try to focus on different types of global-level structural properties), we now believe that there is enough data to adequately evaluate the capability of idpGAN to model global structural properties of the ensembles.

[3] Where is the comparison of the MD simulation data that the DNNs were trained with (i.e., the training set) and the data that the DNN generated (i.e., the output)? Can the authors prove that the DNN did not just memorize the training data? Many readers will be interested in this.

This is an important point. We did evaluate idpGAN on sequences not included in the training data. However, those sequences could be similar to the sequences in the training set, or the model could have learned primarily generic features that apply to all sequences, whether in the training set or not.

To further address this point, we evaluated an idpGAN model on proteins present in its training set and compare its performance with idpGAN models that did not use these proteins in training. We found that the level of performance is similar. This leads us to conclude that idpGAN does not significantly suffer from overfitting on its training data. This is explained in a paragraph that we added at the end of the “***IdpGAN based on CG simulations***” section and in **Supplementary Figure 13**.

Moreover, to provide more direct evidence that idpGAN is not memorizing training set conformations, we also performed a nearest neighbor search of the generated conformations over the whole training set. As a control, we performed similar searches using MD conformations from test set proteins (which are not present in the training set). If idpGAN would memorize training data, its conformations would be more similar to the training set conformations than the MD-generated conformations for the test set proteins. On average, we found comparable levels of similarity between idpGAN and the MD data, suggesting that idpGAN does not directly memorize the geometries from the training data. This is explained in the final paragraph of the “***IdpGAN based on CG simulations***” section and in **Supplementary Figure 14**.

[4] Did the authors use the model trained on one molecule to sample the conformation space of another molecule? If not, why not? If yes, where is this information in the manuscript? There should be a vivid way of conveying this to the readers. I understand the authors hinted at this in the introduction, but I do not know if this is clearly shown in specific display items.

In the original manuscript, we tested idpGAN on proteins not present in its training set. For the C α -based CG ensemble modeling task we used the *IDP_test* set of sequences. This information is reported at the beginning of the “***IdpGAN based on CG simulations***” results section and in the “***Training and test sets for the CG-based idpGAN***” methods section. To make this clearer to readers, in the former section we modified this phrase:

“we evaluated our model on CG MD data for a set of 31 selected IDPs, named IDP_test, that have no similar sequences in it”

to:

“we evaluated our model on CG MD data for a set of 31 selected IDPs, named IDP_test, that are not in the training set and have no similar sequences in it”.

We also added a similar phrase in the **Abstract**.

For the newly-introduced ABSINTH ensemble modeling task, we used the *ABS_test* set of sequences. This is explained in the **“IdpGAN based on all-atom implicit solvent simulations”** results section and the **“Modeling the C α trace of all-atom implicit solvent trajectories”** method section.

[5] What is the largest protein that the model was tested on? Did the authors test the approach on larger and more realistic systems (such as proteins and complexes with 200 to 500 residues) that are often the subject of most MD simulations?

The maximum length that we tested for the C α -based CG modeling task is 200 residues, even although the maximum length used in idpGAN training was 110 residues. In order to extend idpGAN to larger proteins it is necessary to generate additional training data and expand the capacity of our neural networks. This would incur significant computational challenges. We do not believe that there are fundamental limitations in expanding our approach to larger systems and this will be a focus of future efforts. As for modeling IDPs, we believe that many systems of interest have less than 200 residues and therefore our current method should already be useful.

[6] How did the authors conclude that idpGAN is efficient and requires negligible computational cost? Vividly address this in the manuscript if not already done, and make it more obvious.

To arrive at this conclusion in the original manuscript, we timed idpGAN computational wall clock time when generating increasing amounts of conformations (see **Supplementary Figure 24**). To make a comparison with MD computational efficiency, in the original manuscript we also measured the time it takes for idpGAN to generate enough conformations to obtain the highest possible *KLD_r* score with a reference long MD simulation (this score evaluates the divergence of radius-of-gyration distributions). We then measured the time it takes for a MD simulation to reach the same *KLD_r* score. We found that MD takes order of magnitude more time to obtain *KLD_r* score higher than the idpGAN one. This is explained in the **“IdpGAN sampling speed”** results section and in the **“Evaluating sampling efficiency”** methods section in the original manuscript.

To make this clearer to readers, we specified in the former section that we are measuring actual computational “wall clock times” in these experiments.

[7] How did the authors obtain the parameters for the CG models? Although the authors listed the parameter in the “Coarse-grained molecular dynamics simulations” section, it is unclear how you obtained the parameters. For example, how did the authors obtain the value for the spring constant, the equilibrium bond length, etc.? These should be clearly stated, so it is effortless for the readers to find.

We describe the CG model in a separate paper currently under review. A preprint is available on bioRxiv: <https://doi.org/10.1101/2022.08.19.504518>

We added a reference to the bioRxiv paper which can be updated if the CG model paper is accepted.

[8] Did the authors use idpGAN for protein complexes (i.e., for protein systems containing more than one chain of protein)? If not, why not?

We did not test idpGAN on protein complexes, but this is an excellent challenge for future work. We did not pursue it here for two reasons: the first one is a lack of training data. Many IDPs exert their biological role in cells by establishing protein-protein interactions (PPIs) and these interactions involve disorder-to-order transitions of the IDPs. Our C α -based CG model was not parametrized to describe such type of transitions. Since we cannot simulate these PPI interactions with our C α -based model, it makes less sense to train idpGAN to model them. In principle, one could run atomistic simulations of systems involving PPI interactions and train a generative model to sample from the conformational ensembles of unseen protein complexes. However, this presents a formidable computational challenge, because of the large size and the enormous quantity of systems to simulate (given the high number of PPI interactions that exists in cells). A second reason is that adding interactions between chains increases the complexity of the generator network. To our knowledge, no generative modeling study has approached this challenge so far and it will take some effort to determine suitable architectures that can model ensembles of IDPs when interacting with other molecules.

We added some discussion on this topic at the end of the revised manuscript.

REVIEWERS' COMMENTS

Reviewer #1 (Remarks to the Author):

I greatly appreciate the authors' strong efforts to address the concerns raised. I think this is an extremely exciting work that represent a key progress in modeling of disordered protein ensembles. The new data on reproducing the atomistic ensembles are quite convincing and clearly demonstrate that idpGAN has the capacity for predicting nontrivial structural features. I am fully supportive of its publication in Nature Communications!

A couple small requests/questions, how ABS_test set was constructed beside the sizes? It may be useful to also show the average residue helicity profiles in addition to the overall helicity alone, which more clearly show if idpGAN can capture the residual structures in IDPs.

Reviewer #2 (Remarks to the Author):

The authors have adequately addressed all previous comments, and the manuscript is now in a much better condition. I would recommend its acceptance for publication.

We sincerely thank the reviewers for their revision work and truly appreciate their comments on the revised manuscript. A response to the remaining concerns of Reviewer #1 is given below:

Reviewer #1

A couple small requests/questions, how ABS_test set was constructed beside the sizes?

The peptides in the ABS_test set are intrinsically disordered peptides for which ABSINTH simulations were previously performed (see the Mao et al. 2010 reference in the main text of the manuscript) and simulation results were found to be in close agreement with experimental data. Therefore, we could test whether idpGAN can recapitulate: (a) already published simulation results, (b) experimental observables and biophysically relevant properties of these peptides. Additionally, none of the ABS_test peptides share sequence similarity (according to the definition presented in our “Methods” section) with the training set sequences and so they could be readily used as a test set.

We added this information in the main text of the manuscript.

It may be useful to also show the average residue helicity profiles in addition to the overall helicity alone, which more clearly show if idpGAN can capture the residual structures in IDPs.

We followed the advice and added a figure (Supplementary Figure 20) which shows residue-level helicity profiles. For most of the peptides, idpGAN can identify the regions with higher tendencies of assuming helical geometries.